# Possible influence of Sudden Stratospheric Warmings on the atmospheric environment in the Beijing-Tianjin-Hebei region

Qian Lu[1,2,3], Jian Rao[1,*], Chunhua Shi[1], Dong Guo[1], Guiqin Fu[2], Ji Wang[4], Zhuoqi Liang[1]

[1]Key Laboratory of Meteorological Disaster, Ministry of Education (KLME) / Joint International Research Laboratory of Climate and Environment Change (ILCEC) / Collaborative Innovation Center on Forecast and Evaluation of Meteorological Disasters (CIC-FEMD), Nanjing University of Information Science and Technology, Nanjing 210044, China
[2]Key Laboratory of Meteorology and Ecological Environment of Hebei Province, Shijiazhuang 050021, China
[3]Chengde Meteorological Service of Hebei Province, Chengde, Hebei 067000, China
[4]Beijing Regional Climate Center, Beijing 100089, China

*Correspondence to*: Jian Rao (raojian@nuist.edu.cn)

**Abstract.** Using the ERA5 and MERRA2 reanalysis, and surface meteorological observation data, this study explores the possible impact of the sudden stratospheric warming (SSW) events on air quality in the Beijing-Tianjin-Hebei (BTH) region. Major SSW events are divided into polar vortex displacement SSW and polar vortex split SSW. As the duration of split SSW events is longer and the stratospheric signal pulses propagate further downward than displacement SSWs, subseasonal variability of the atmospheric particulates in the BTH is larger during split SSWs. The air particulate concentration is light before the SSW onset due to the enhanced perturbation in the troposphere associated with strengthened planetary waves. The air particulate concentration around the SSW onset dates begins to rise due to weakening of the tropospheric disturbance as the enhanced planetary waves enter the stratosphere. In the decaying period of the SSW, the air particulate concentration decreases as the stratospheric negative Northern Annular Mode (NAM) signal propagates downward. Specifically, in the pre-SSW period of displacement (split) SSW events, a wavenumber-1-like (wavenumber-2-like) anomaly pattern is strengthened. The East Asian winter monsoon intensifies as the east Asian trough is deepened especially before the split SSW event onset, leading to a cleaning period. Around the SSW onset period as the tropospheric perturbation diminishes and the East Asian winter monsoon weakens, a surge of air particulate concentration is observed. After the SSW onset, due to the downward propagation of the stratospheric negative NAM signal, cold anomalies form in the northeastern East Asia especially for split SSWs, corresponding to a cleaning period in the BHT region. The local meteorological conditions during the SSWs are also discussed.

**Key words**: Sudden stratospheric warming (SSW), Beijing-Tianjin-Hebei (BTH), Atmospheric environment, East Asian winter monsoon

## 1. Introduction

Haze pollution is reported to cause serious adverse effects on the ecological environment, daily life and transportation, damage the health of humans and animals, and even reduce the yield of crops (Li et al., 2014; Hu et al., 2015; Chen et al.,

2020; Bu et al., 2021). Considering the wide influence of the atmospheric particulate, haze has become a hot topic attracting widespread concern, although the air quality has improved in the past decade (Ding et al., 2019; Li et al., 2019). The source analysis of pollutants shows that the primary local emission from factories and automobiles, atmospheric transport from

35 surrounding regions, and secondary formation of aerosols by chemistry reactions produce the main atmospheric haze pollution (Huang et al., 2015; Zhao et al., 2019). Adverse meteorological conditions can lead to fast accumulation or slow dilution and difficult diffusion of pollutants in the atmosphere, which are also the background condition of heavy pollution weather (Li et al., 2018; Dang and Liao, 2019; Chang et al., 2020). Dormancy of the east Asian winter monsoon and the rise of winter temperature are possibly accountable for changes of haze days across North China (Yin et al., 2015; Li et al., 2016).

On the hemispheric scale, the wintertime heavy haze in parts of North China often occurs when the Arctic Oscillation (AO) is in the positive phase (Yin et al., 2017; Li et al., 2018). During the AO positive phase, the low pressure in the Arctic region deepens, and the high pressure in the midlatitudes intensifies, limiting the southward expansion of the cold air in the polar region. As the key system of haze pollution in North China, at times an anomalous anticyclone developed in the Bohai Sea, which is usually accompanied by air quality decline (Wu et al., 2017). The large-scale atmospheric circulation and

meteorological conditions provide a background for the occurrence and decay of heavy haze days (Wang and Chen, 2016; Li et al., 2018). For example, certain conditions are not conducive to the dilution and diffusion of air pollutants, such as the reduction of the local boundary layer height, increase in the static stability (or even development of inversion layer), and deceleration of the near surface wind speed (Huang et al., 2018, 2020; Yang et al., 2016). With those conditions, higher relative humidity might promote the moisture absorption and growth for air particulates (Feng et al., 2018).

Sudden stratospheric warming (SSW) is a rapid and violent warming phenomenon in the polar stratosphere in winter (Charlton and Polvani, 2007; Butler et al., 2015; Rao et al., 2018; Lu et al., 2021a, 2021b). The SSW event is a typical phenomenon of two-way coupling between stratosphere and troposphere (Hu et al., 2014; Rao et al., 2021a). When an SSW occurs, the atmospheric temperature in the polar cap region of the stratosphere suddenly rises, which can increase by tens of degrees Celsius within a week (Charlton and Polvani, 2007; Rao and Garfinkel, 2020; Lu et al., 2021b). After the SSW onset,

the stratospheric anomalies can propagate downward, and the tropospheric circulation is adjusted accordingly (Baldwin et al., 2001; Rao et al., 2021b). If the meridional gradient of the temperature between 60 and 90°N reverses, and the circumpolar westerly winds at 10 hPa and 60°N are significantly weakened but do not reverse to the easterly winds, the SSW is usually classified as a minor event, which is excluded from our analysis. If the circumpolar westerly winds reverse to the easterly winds at 10 hPa and 60°N, it is called a major SSW (Charlton and Polvani, 2007; Rao et al., 2018; Butler et al., 2020;

Baldwin et al., 2021). According to the morphology of the polar vortex shape around the onset time, major SSWs can be further categorized as polar vortex displacement SSW events and polar vortex split SSW events (Charlton and Polvani, 2007; Baldwin et al., 2021). Vortex displacement SSWs are associated with enhanced planetary wavenumber 1, while vortex split SSWs are alternatively forced by wavenumber 1 and wavenumber 2 (Karpechko et al., 2018; Liu et al., 2019; Rao et al., 2019a, 2020; Baldwin et al., 2021). On average, six or seven SSW events occur in a decade, and it can appear in months

from November to March, with the most concentrated from January to February (Cao et al., 2019; Liu et al., 2019; Rao and

Garfinkel, 2021). The SSW appear unevenly in every decade and exhibits a significant interdecadal variation (Rao et al., 2021b). Before the SSW onset for some events, the upward propagation of planetary waves from the troposphere to the stratosphere is enhanced (de La Cámara et al., 2019; Rao et al., 2019b), which might be owing to the preceding tropospheric blocking and/or deepening of the climatological trough (Rao et al., 2018, 2020; Baldwin et al., 2021). Another trigger for

SSWs is the stratospheric dynamics and the vortex geometry in the lowest stratosphere (de La Cámara et al., 2019). After the SSW onset, the atmospheric zonal mean anomalies generated by SSW events in the stratosphere can propagate downward to the troposphere and affect the weather and climate in the troposphere (Lu et al., 2021b; Baldwin et al., 2021). The weakened stratospheric polar vortex during SSW events is mainly projected onto the negative phase of the northern annular mode (NAM). With the negative NAM signal descending to lower levels, the tropospheric anomaly circulation evolves into a

pattern resembling the negative phase of the Arctic Oscillation (AO) or the negative phase of the North Atlantic Oscillation (NAO) in the Atlantic sector (Baldwin and Dunkerton, 1999, 2001). On the probabilistic sense, cold air outbreak is likely to increase on the northern continents (Yu et al., 2015; Lu et al., 2021b; Rao et al., 2021a). Although most previous studies emphasized the possible impact of the stratospheric disturbance on the tropospheric circulation, the possible linkage between the stratospheric anomalies and the subseasonal variability of the regional air quality is still not well explored.

Existing evidence mainly focuses on the possible impact of tropospheric climate anomalies and the wave train like teleconnections on the regional air pollutions, but few studies investigate the possible impact of stratospheric changes on regional haze pollution. Given that the tropospheric climate anomalies can be affected by the stratospheric changes, the haze pollution in the Beijing-Tianjin-Hebei (BTH) region associated with atmospheric circulation variability may also be affected by the stratospheric changes (Chang et al., 2020; Huang et al., 2021; Lu et al., 2021a, 2022). A recent study reports that the

weakening of stratospheric polar vortex in the winter of 2015/16 can lead to easier diffusion of pollutants and a gradual improvement of the air quality in the BTH region (Huang et al., 2021). Comparing the three SSW events in February 2018, January 2019 and January 2021, it is shown that the subseasonal variability of the $PM_{2.5}$ (small particles with the aerodynamic diameter equal to or less than 2.5 μm in the atmosphere) concentration in the BTH region might be enhanced after the stratospheric anomalies propagate downward to the troposphere and near surface (Lu et al., 2021a, 2022). However,

the generalization of the possible impact of the SSWs on the BHT region in previous studies should be verified using abundant historical SSW samples. This study will adopt more SSW samples and robustly establish the relationship between the SSW and the air particulate concentration in the BHT region. Hitherto, we still do not clearly know yet if the vortex displacement and split SSWs have similar or different impacts on the regional atmospheric environment in the BTH region. The paper is constructed as follows. Following this part, the data and methodology are briefly described in section 2. Section

3 shows the composite of the zonal-mean circulation anomalies for displacement SSW events and split SSW events. Evolutions of large-scale circulation anomalies accounting for the subseasonal variability of the atmospheric particulates in the BHT region during different periods of the two types of SSWs are analyzed in section 4. Section 5 analyzes the local meteorological anomalies in the BTH region during different periods of the two types of SSWs. Section 6 display the

composite atmospheric environment quality in the BTH region using different metrics during the two types of SSWs. Finally, section 7 presents summary and discussion.

## 2. Data and methodology

Daily reanalysis data from 1980 to 2021 is provided by the European Center for Medium Range Weather Forecasts, and its fifth-generation reanalysis is used (ERA5) (Hersbach et al., 2020). The atmospheric data used in this study include the geopotential height ($Z$, geopotential divided by 9.8 m s$^{-2}$), the zonal wind ($U$), the meridional wind ($V$) and air temperature ($T$). This reanalysis was downloaded at a 1° × 1° horizontal resolution at 37 pressure levels spanning from 1000 hPa to 1 hPa. The ERA5 surface data employed in this study include the sea level pressure (SLP), surface air temperate (SAT) and planetary boundary layer height (PBLH), which is also collected at a 1° × 1° horizontal resolution. The daily observation data of minimum visibility, haze, fog and light fog in the cities of Beijing, Tianjin and Shijiazhuang are provided by the China Meteorological Information Center (http://data.cma.cn/). The second Modern-Era Retrospective analysis for Research and Applications (MERRA-2) is also used, which has a horizontal resolution of 0.625°×0.5° (longitude × latitude). The MERRA2 reanalysis is provided by the NASA and begins from 1980 (Gelaro et al., 2017). The aerosol optical depth (AOD) data from the MERRA2 are used to denote the historical atmospheric environment conditions in the BTH region when the PM$_{2.5}$ concentration was still not a standard observation variable in China. The long-term mean from 1980–2021 on each calendar day of the year is calculated to denote the raw daily climatology. Daily anomalies refer to the detrended deviation from this smoothed daily climatology with a window of 91 days (three months or one season) to remove the high-frequency variability. The results are unchanged if we change the window between 61 and 121 days.

The modified World Meteorological Organization (WMO) SSW identification algorithm is used to select major SSW samples and their onset date. When the circumpolar westerly winds at 60°N and 10 hPa reverse to easterly wind and last for at least 5 days (removing some marginal SSWs), an SSW sample is determined. The first day on which the zonal mean zonal winds reverse is defined as the SSW onset date. According to the position and shape of polar vortex, major SSW events can be further divided into two types, polar vortex displacement type and polar vortex split type (Charlton and Polvani, 2007; Butler et al., 2015; Rao et al., 2021b). Vortex-centric diagnostics are used to categorize the type of SSW events, which can calculate the vortex centroid latitude and longitude (Seviour et al., 2016). In addition to the vortex-centric parameters, the aspect ratio can also be calculated based on the two dimensional vortex moment diagnosis of the vortex shape, which are used to define a vortex uniquely, and an "equivalent ellipse" is defined as the representative of a vortex (Mitchell et al., 2011; Seviour et al., 2016). For a simplified purpose, the geopotential height is used to determine the absolute vortex moments ($M_{ab}$) and the relative vortex moments ($J_{ab}$), although similar procedures can also be applied to the Ertel's potential vorticity. The geopotential height is projected onto a Cartesian coordinate first, and the Arctic polar stereographic projection is used. Based on this vortex moment diagnosis, several parameters of the stratospheric polar vortex at 10 hPa are finally determined, including the vortex centroid x-coordinate value, the vortex centroid y-coordinate value, the vortex area in the Cartesian

coordinate, and the aspect ratio between the major and minor axes of the equivalent ellipse (Matthewman et al., 2009). Finally, the coordinates of the vortex centroid are conversed to the spheric coordinate, and the latitude and longitude of the vortex centroid are obtained. With the evolution of those metrics during the SSW, a split SSW should meet the requirement that the aspect ratio is larger than 2.4 for at least seven days during the period from 10 days before to 10 days after the SSW onset date. A displacement SSW can be confirmed if the vortex centroid latitude is in the equatorward side of a latitude threshold (i.e., 66°N) for at least seven days. Several recent studies reported that this methods can select similar SSW samples as the classification using conventional methods (Seviour et al., 2016; Cao et al., 2019; Rao et al., 2021b). One-sample two-sided $t$-test is to test whether the difference between a sample average and a known population average is significant. When the population distribution is normal, but the samples are not large, the deviation of the sample mean from the population mean show a $t$-distribution. As the SSW samples are limited, it is reasonably to use the Student's $t$-test for this study. The one-sample $t$-test is calculated as $t = \frac{\bar{x}-\mu}{\frac{\sigma_x}{\sqrt{n}}}$, where $n$ is the sample number, $\bar{x}$ is the sample mean, and $\sigma_x$ is the sample standard deviation. The null hypothesis is that the $t$-value is zero (i.e., $\bar{x}=\mu$) if the sample mean shows insignificant difference from the population mean. Otherwise, the sample mean is significant different from the population mean if the null hypothesis is rejected (Krzywinski and Altman, 2013). In order to test the credibility and consistency of data, the bootstrap method is adopted to calculate the confidence level (e.g., Alfons et al., 2022) for the mean visibility, haze days, fog days, light fog days by resampling 1000 times with a sample size proportion of 0.5 for both displacement and split SSWs.

## 3. Evolutions of the zonal mean circulation anomalies during two types of SSWs

According to the WMO algorithms, 17 major SSW events occurred in the Arctic stratosphere during the winter of 1981–2021. Some marginal SSWs with a small deceleration of the zonal winds or with one or two days of easterlies have been removed from our composite with a requirement of wind reversals for at least five days (e.g., Hu et al., 2014). There are 8 vortex displacement SSW events and their onset dates are as follows: 24 January 1987, 15 December 1998, 31 December 2001, 5 January 2004, 21 January 2006, 24 February 2007, 22 February 2008, and 5 January 2021. There are 9 vortex split SSW events and their onset dates are as follows: 2 January 1985, 10 December 1987, 24 February 1989, 12 February 2001, 27 January 2009, 12 February 2010, 10 January 2013, 11 February 2018, and 2 January 2019. Using the vortex-centric diagnostics and removing the marginal events, it can be found that the vortex displacement SSWs and vortex split SSWs in midwinter is comparable in their numbers during 1980–near present.

The composite pressure-time evolution of the polar cap (i.e., area-averaged over 60–90°N) temperature anomalies, zonal-mean zonal wind anomalies at 60°N, and polar cap height anomalies from day -30 to day 50 relative to the onset date of the two types of SSWs are shown in Fig. 1. When the displacement SSWs are considered, the zonal mean zonal wind anomalies at 60°N only appear above 200 hPa and do not propagate downward to lower troposphere. The easterly anomalies begin to appear 10 days before the onset dates and last until day 35. However, only the easterly anomalies above 50 hPa from day -10

to day 20 are statistically significant and reach the maximum (-30 m/s) around the onset dates (Fig. 1b). In contrast, the easterly anomalies begin to appear since day -15 for split SSWs, and the wind anomaly magnitude is also stronger, reaching the maximum intensity (-35 m/s) several days before the SSW onset (Fig. 1e). The easterly anomaly signal can last until day

50, which propagate downward to the near surface since day 20. However, only the easterly anomalies above 200 hPa from day -15 to day 40 are statistically significant. A comparative analysis of the zonal mean zonal wind anomalies for the displacement and split SSWs, it is revealed that the easterly anomalies for split SSWs lasts longer with the anomaly magnitude for the latter larger than the former. The easterly anomalies propagate more downward to the troposphere during the split SSWs than displacement SSWs. The composite results are consistent with the case studies: the circulation anomalies

during the February 2018 SSW are more easily propagate downward to the troposphere than the January 2019 SSW (Rao et al., 2020; Lu et al., 2021a).

According to the principle of thermal winds, the weakening of the 60°N zonal westerly wind is dynamically associated with the decrease and even reversal of the meridional temperature gradient between midlatitudes and the North Pole. When displacement SSWs are considered, warm anomalies appear since day -23 to day 30 and the warmest anomalies (~20℃)

appear on day -5, implying a sudden rise of the Arctic stratospheric temperature. The warm anomalies do not show a significant downward propagation and mainly develop above 200 hPa (Fig. 1a). Similarly, the positive temperature anomalies begin to appear since day -20 for the split SSWs and reach the maximum value of 15℃ on day -8. Warm anomalies diminish gradually in the stratosphere and persisted until day 30 (Fig. 1d). However, it can still be observed that the warm anomalies during split SSWs propagate deeper downward, and warm anomalies around 10 ℃ during split SSWs

spread below 100 hPa. No significant zonal-mean temperature anomalies are observed in the troposphere, which might be due to the nonuniform response of the temperature anomalies in the troposphere and the interlocking distribution of warm and cold anomalies at the same latitudes (Liu et al., 2019; Lu et al., 2021a; Rao et al., 2021a; Liang et al., 2022).

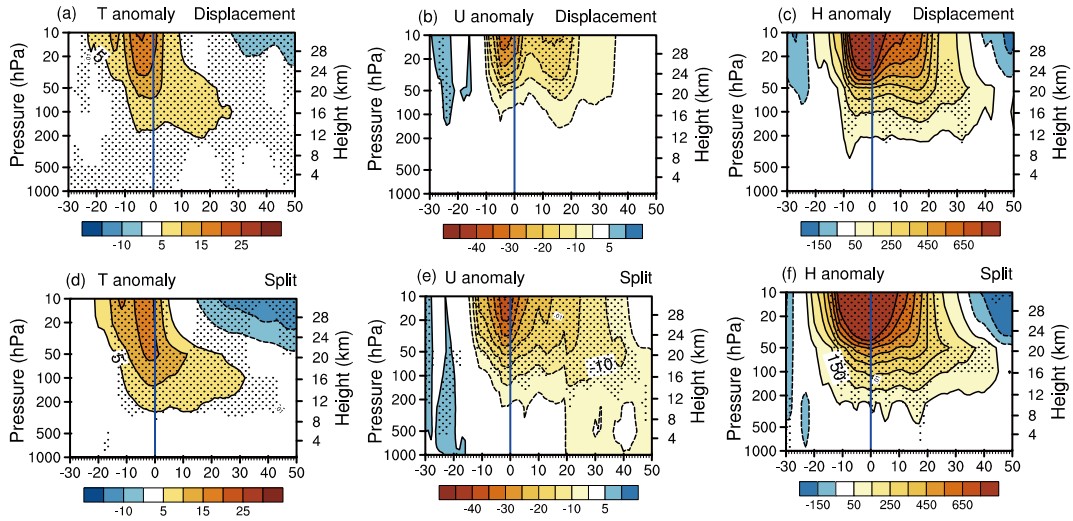

**Figure 1: Composite pressure-temporal evolution of the SSW from day -30 to day 50 relative to the onset date for vortex displacement SSWs (top row) and vortex split SSWs (bottom row). (a,d) Temperature anomalies over the polar cap from 60°-90°N (interval: 5; units: °C). (b,e) Zonal-mean zonal winds at 60°N (interval: 5; units: m s⁻¹). (c,f) Geopotential height anomalies over the polar cap from 60°–90°N (interval: 200; units: gpm). The onset date of the SSW is marked by a vertical blue line. Black dots indicate the composite anomalies at the 95% confidence level using the *t*-test.**

The zonal-mean circulation response to SSWs is also shown for the polar cap heights. According to the principle of geostrophic winds, decelerated westerly jets in the circumpolar region are accompanied with rise of polar cap height and/or the reduce of the midlatitude heights, and accelerated winds are accompanied with decrease of polar height and/or rise of midlatitude heights. Positive geopotential height anomalies are consistent with development of the negative stratospheric NAM during both displacement and split SSWs (Fig. 1c, f). For displacement SSW events, the positive geopotential height anomalies begin to appear around day -20 and reach the maximum (~750 gpm) on day -8 at 10 hPa. The negative NAM response gradually weakens after the SSW onset and persists until day 45, and meanwhile the positive height anomalies propagate downward. The positive height anomalies mainly develop in the stratosphere, and exhibit a downward propagation to the lowest level on day -10, but after the SSW onset the negative NAM-like signal can descend deeper to 200 hPa (Fig. 1c). In contrast, for the split SSWs, the positive geopotential height anomalies appear since day -22 and propagate downward to the troposphere instantly, where significant signals are observed. The maximum positive height response appears on day -12 at 10 hPa (~850 gpm), and afterward the height anomaly magnitude gradually weakens with significant anomalies lasting until day 45 (Fig. 1f). The stratospheric positive geopotential height anomalies can propagate to the troposphere since day 18. In short, the composite of different variables seems to denote that the intensity of the vortex split SSW events is on average stronger than vortex displacement SSWs.

The stratospheric circulation changes drastically during the SSW, which in turn has a downward influence on the tropospheric circulation anomalies (Baldwin et al., 2021 and references therein). By exploring the tropospheric circulation variability associated with the two types of SSWs, the role of the stratosphere in modulating the regional air environment might be better understood.

## 4. Large scale circulation anomalies in the stratosphere and troposphere

### 4.1. Evolution of stratospheric circulation anomalies

SSW is a typical representative of the stratosphere-troposphere coupling in the extratropics, which have a downward influence on the tropospheric circulation anomalies (Lu et al., 2021a, 2022). The large circulation anomalies in the stratosphere might further affect the local meteorological conditions for air particulate diffusion in the BTH region. Figure 2 shows the evolution of 10 hPa geopotential height anomalies in the Northern Hemisphere during vortex displacement and split SSW events. In the pre-SSW period (P1) the geopotential height anomalies distribute as a wavenumber-1-like pattern, and the negative height center is located around the Nova Zembla, and the significant positive height center is located around the Bering Strait and North Pacific (Fig. 2a). Around the onset date of displacement SSWs and afterward (P2), the polar

vortex weakens rapidly. Meanwhile, the North Pole is occupied by large positive height anomalies, and the maximum height anomaly exceeds 1000 gpm at the 95% confidence level. The mid-to-low latitudes are covered by weak negative height anomalies (Fig. 2b). This zonally uniform positive anomalies between middle and high latitudes denotes the development of the negative NAM. After the SSW onset date for displacement events (P3), the Arctic region is still occupied by the positive height anomalies, but the significant anomaly magnitude weakens and the coverage shrinks as compared with the preceding period (Fig. 2c).

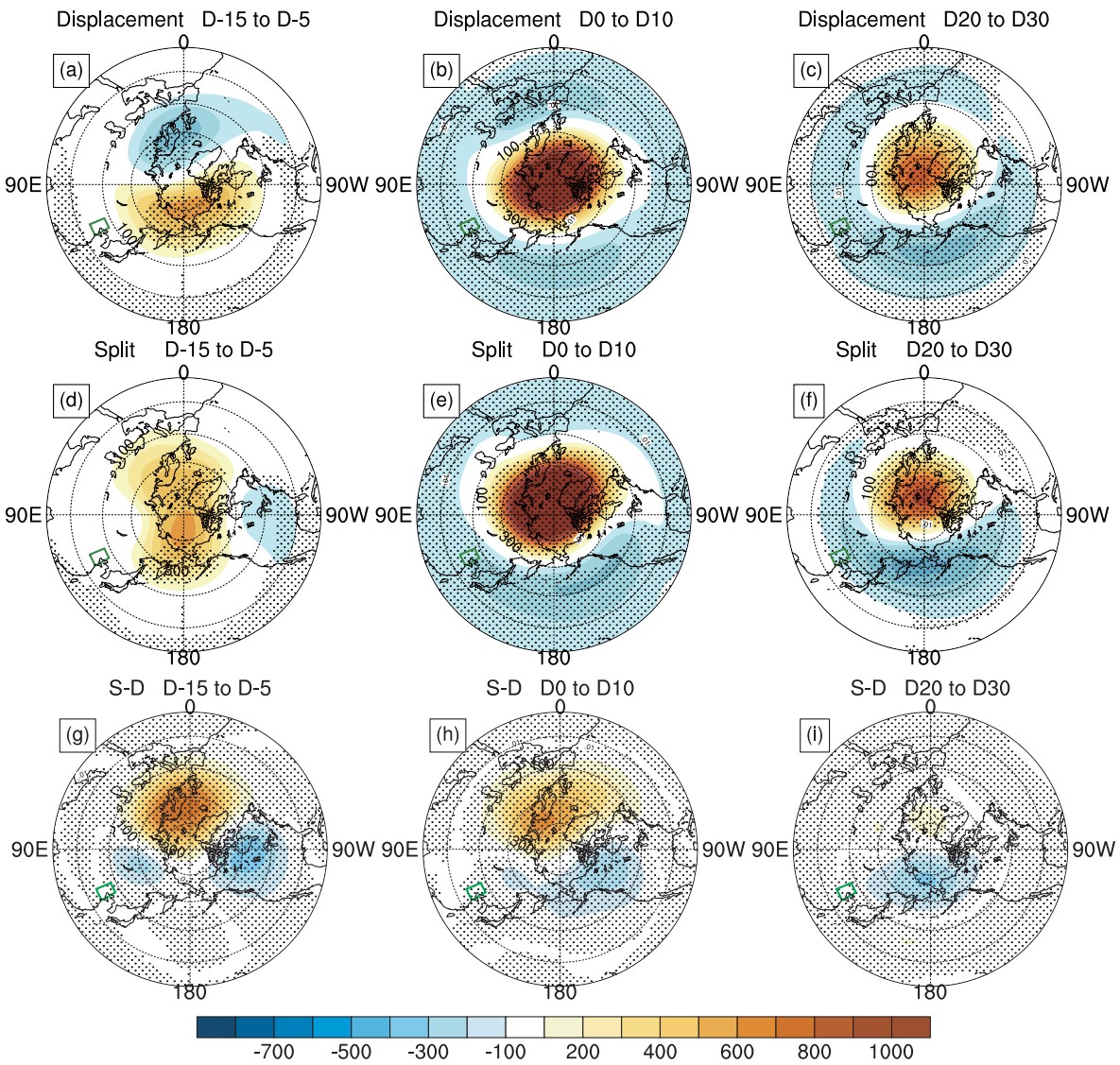

**Figure 2: Composite geopotential height anomalies in the Northern Hemisphere at 10 hPa during three periods of the SSW for vortex displacement events (top row) and vortex split events (bottom row). (a, d) Day -15 to day -5 in the pre-SSW period. (b, e) Day 0 to day 10 in the SSW onset period and afterwards. (c, f) Day 20 to day 30 in the SSW decaying period. (g, h, i) Composite difference in the geopotential height anomalies between split and displacement SSWs. The dotted regions mark the composite**

In the pre-SSW period for split events (P1), the polar vortex weakens as positive height anomalies occupy most of the Arctic (Fig. 2d). The positive height anomaly elongated in two directions, one toward the Nova Zembla, and the other toward the Bering Strait and the North Pacific with a narrow band crossing the North Pole covered with positive height anomalies. Those two positive anomaly centers are significant at the 95% confidence level. This elongation of the positive anomalies is probably related to the dynamical role of the wavenumber-2. Weak negative height anomalies are seen over the Great Lakes, implying the vortex is far biased from the Arctic. In the SSW onset period (P2) for events, a negative NAM-like pattern forms: similar to the circulation change for displacement SSWs, the Arctic is occupied by significant positive height anomalies with the maximum exceeding 1000 gpm (Fig. 2e). In the SSW decaying period (P3) for split events, the negative NAM-like pattern gradually weakens (Fig. 2f).

In the SSW onset and decaying periods for both displacement and split SSWs, a negative NAM is observed. To clearly reveal the difference between the split and displacement SSWs, the split minus displacement composite is also shown (Fig. 2g–i). The composite in the pre-SSW period shows that the difference is largest and most significant in the North Atlantic (Fig. 2g). In the SSW onset period, this composite difference resembles a wavenumber-1 like pattern, which denotes a stronger wave-1 forcing for displacements than splits (Fig. 2h). In the post-SSW period, the difference is still evident over the Bering Strait and North Pacific (Fig. 2i).

In short, displacement and split SSW events show somewhat differences in the stratospheric circulation anomalies before the SSW onset. The displacement SSWs is preceded by a wavenumber-1-like circulation pattern, while the split SSWs is preceded by a wavenumber-2-like circulation pattern. In the SSW onset and decaying periods for both displacement and split SSWs, a negative NAM is observed. The positive height center over the Arctic for split SSWs is more inclined to the Iceland and Greenland, whereas this center for displacement SSWs is round over the North Pole. Since the stratospheric NAM signal can propagate downward to the troposphere, which might affect the tropospheric meteorological conditions, we will examine the tropospheric evolutions.

## 4.2. Evolution of mid-tropospheric circulation anomalies

Figure 3 shows evolutions of the geopotential height anomalies at 500 hPa for displacement and split SSWs. Before the displacement SSW onset (P1), the tropospheric height anomalies also present wavenumber 1 pattern. In the Pacific sector, the anomalous low center is located over the Aleutian Islands, and the anomalous high center is located over subtropical Pacific. In the Atlantic sector, significant positive height anomalies develop over the Greenland and North Atlantic (Fig. 3a). The general distribution resembles the negative Pacific–North American pattern (PNA), which amplifies the climatological wavenumber 1. The strengthening wavenumber 1 can propagate upward to disturb the stratosphere and excite a displacement SSW event. In the SSW onset period (P2) for displacement events, positive anomalies occur in the Arctic, negative height anomalies occur in northern Eurasia and northern North America at high latitudes. Meanwhile, two significant negative

anomaly centers appear over the North Pacific, and positive height anomalies occur in western Eurasia at midlatitudes. (Fig. 3b). After the displacement SSW onset (P3), large and significant negative height anomalies appear over Nova Zembla extending westward, while positive height anomalies form in Greenland and the Arctic, resembling the negative phase of the NAO (Fig. 3c). In the Pacific sector, an evident strong positive height anomaly center appears over the North Pacific, and negative anomalies appear over the western coasts of North America, implying the phase conversion of the PNA teleconnection from positive to negative.

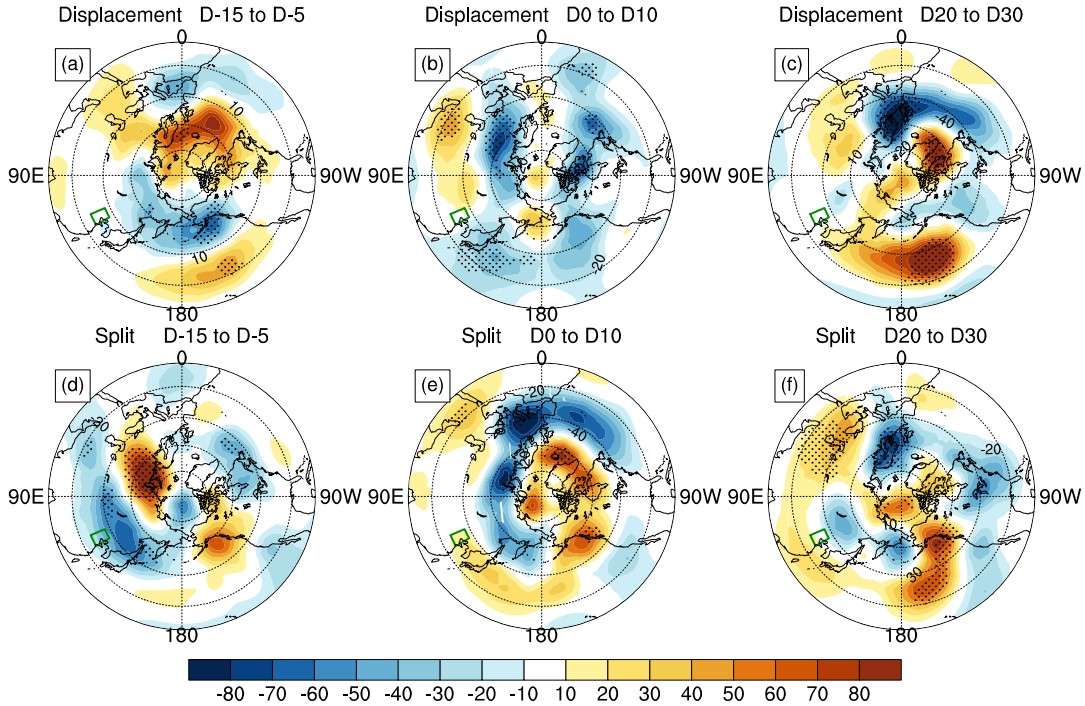

**Figure 3: As in Figure 2 but for composite geopotential height anomalies in the Northern Hemisphere at 500 hPa during three periods of the SSW for vortex displacement events (top row) and vortex split events (bottom row). The dotted regions mark the composite geopotential height anomalies at the 95% confidence level based on the two-sided Student's *t*-test. The green box marks the focused BTH region.**

In the pre-SSW period for split SSW (P1), the geopotential height anomalies present a wavenumber-2 pattern (Fig. 3d). Two significant anomalous low centers are located in eastern Siberia and eastern Canada, respectively, while two anomalous high centers (one with the 95% confidence level) are correspondingly located around the Urals and Alaska. This height anomaly pattern is in phase with and therefore amplifies the climatological wavenumber 2. The enhanced wavenumber 2 can also propagate upward into the stratosphere and split the polar vortex (Rao et al., 2018; Baldwin et al., 2021). In the SSW onset period (P2) for split events, positive height anomalies occur in the Arctic Canada, Greenland, Iceland, and the midlatitude Pacific, while negative height anomalies appear in northern Eurasia and the midlatitude Atlantic (Fig. 3e). In the Atlantic sector, a negative NAO-like anomaly pattern generates. In the SSW decaying period (P3) for split SSWs, the negative NAO-

280 like anomaly pattern still persists in the Atlantic sector, while the significant anomalous high over the Urals is stronger during this period for split (Fig. 3f) than displacement.

### 4.3. Evolution of the near-surface temperature anomalies

The distribution of the near surface temperature anomalies at a 2-meter height (t2m) during three periods are shown in Fig. 4 for displacement SSWs and split SSWs. Before the SSW onset (P1) for displacement events, warm anomalies appear in the
285 Arctic Canada, Greenland, and most parts in the Central and East US, whereas in the western coasts of the US, cold anomalies are evident (Fig. 4a). This significant temperature pattern is probably related to development of the positive PNA. Cold anomalies develop in the high latitudes of the Eurasian continent, and warm anomalies appear in parts of the southern Eurasia. The temperature anomalies in the BTH region are relatively weak. In the SSW onset period (P2), the cold anomalies in northern Eurasia at high latitudes weaken with its coverage shrank (Fig. 4b). Most parts of the Eurasian and North
American continents are controlled by significant warm anomalies, corresponding to a dormant period of weakening east Asian monsoon. Warm anomalies in North China and weakened east Asian monsoon are consistent with the worsening atmospheric quality (shown later). After the SSW onset (P3) for displacement events, areas covered with warm anomalies further shrink, and cold anomalies are confined to the Arctic, the Bering Strait, and North America (Fig. 4c).

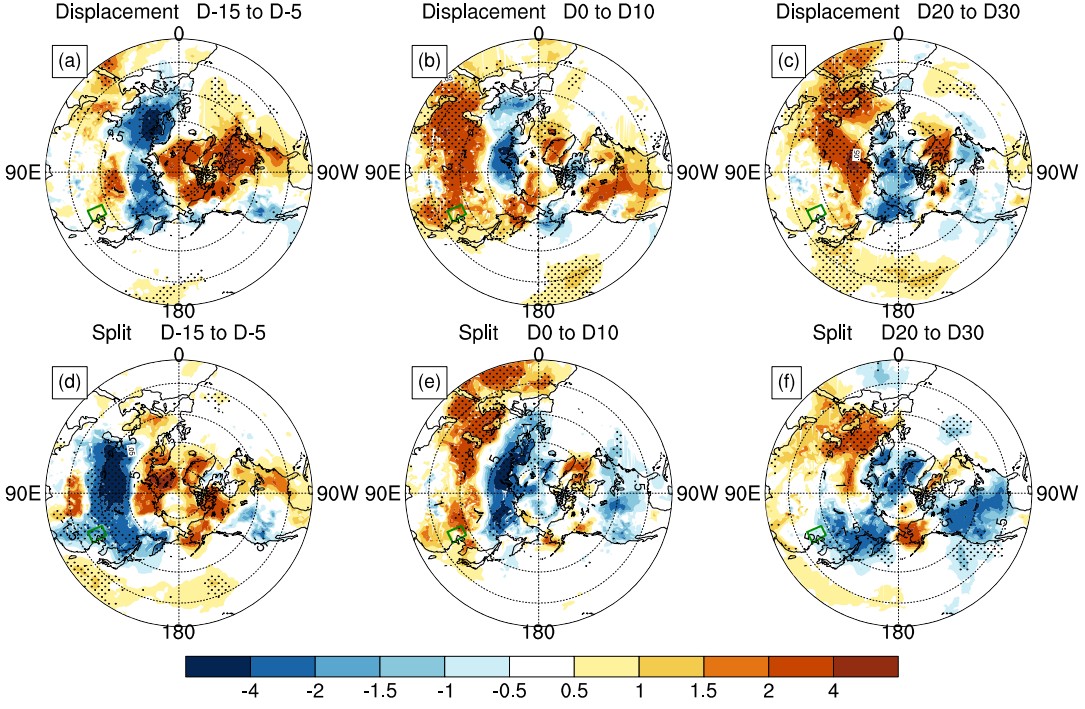

**Figure 4: Composite 2-meter temperature (t2m) anomalies in the Northern Hemisphere during three periods of the SSW for vortex displacement events (top row) and vortex split events (bottom row). (a, d) Day -15 to day -5 in the pre-SSW period. (b, e) Day 0 to day 10 in the SSW onset period and afterwards. (c, f) Day 20 to day 30 in the SSW decaying period. The dotted regions**

**mark the composite t2m anomalies at the 95% confidence level based on the two-sided Student's *t*-test. The green box marks the focused BTH region.**

In the pre-SSW period (P1) for split events, warm anomalies are also observed over the Arctic and the eastern part of the North American continent, while significant large cold anomalies appear in the Eurasian midlatitudes (Fig. 4d), implying strong cold advection in North China denoted by the local cold anomalies. In contrast, the cold anomalies in the western US are much weaker. In the onset period (P2) for split SSW events, high-latitude Eurasia and most part of North America are covered by cold anomalies, while the mid- and low-latitudes of Eurasia are dominated by warm anomalies (Fig. 4e). Warm

anomalies and weakened east Asian winter monsoon are consistent with a period of higher air particulate concentration (shown later). After the split SSW onset (P3), the warm anomalies in the Eurasia weaken with the coverage shrank. Meanwhile, cold anomalies extend to East Asia as the cold anomalies in North America further intensify (Fig. 4f). Cold anomalies possibly denote that the winter monsoon might enhance as the stratospheric signal propagates downward to the troposphere, and a cleaning process of the atmospheric environment in the BTH region happens.

**5. Local meteorological conditions in the BTH region**

During different periods of the SSW, the stratosphere-troposphere coupling is different. In the P1 period, the tropospheric waves propagate upward and begin to disturb the stratosphere. In the P2 period, the stratospheric variation reaches the climax. In the P3 period, the stratospheric signal propagates downward to the lower troposphere and poses a potential impact on the regional environment (Lu et al.,2021a, 2022). Previous studies have shown that the boundary layer height can suitably

measure the diffusion potential for the air particulates (Huang et al., 2018, 2020). The boundary layer height (BLH) anomalies during different periods of the SSW are shown in Fig. 5 for displacement and split SSWs. By comparison, the BLH anomalies are not so evident during displacement events than during split events. In the pre-SSW period for displacement events, the negative height anomaly at 500 hPa in the northeast of Asia denotes a deepening of the local trough, which helps to strengthen the northwesterlies in the BTH region (Fig. 3a), and obvious northwesterly anomalies can also be

observed in the near surface layer (Fig. 5a). Further, warm anomalies appear in the near surface around the BTH region (Fig. 4a), and therefore, negative BLH anomalies occur to the east of BTH region in the pre-SSW period (Fig. 5a). The significant negative anomaly center over the North Pacific to the east of Japan is contrasted with the positive anomaly center over the south of Baikal Lake area, and both of them intensify the northerly wind in the BTH region (Fig. 3b). The northerly wind anomalies develop downward to the near surface layer (Fig. 5b). The surface temperature in BTH region are significantly

warm compared to the previous period (Fig. 4b). The negative BLH anomalies still persist in the Bohai Sea during the SSW onset period, and no simultaneous evident anomalies are found in the BTH region (Fig. 5b). In the post-SSW period for displacement events, the height anomalies at 500 hPa are very weak (Fig. 3c), and near-surface wind anomalies are also fairly small in the BTH region (Fig. 5c). Warm anomalies in the BTH region weaken (Fig. 4c), correspondingly to the weak

and insignificant positive BLH anomalies (Fig. 5c). This is consistent with the relatively weak downward propagation of stratospheric pulse associated with displacement SSWs.

In contrast, the evolution of the BLH anomalies is more dramatic during split SSWs than during displacement SSWs. Specifically, in the pre-SSW period for split events, the negative anomaly center in the northeast of Asia at 500 hPa also strengthens the climatological trough at 500 hPa (Fig. 3d), which favors an enhancement of the northwesterlies in the BTH region. The enhanced northwesterly anomalies can also be clearly observed from the near surface winds (Fig. 5d). In addition, cold anomalies develop in the lower troposphere (Fig. 4d), and therefore significant positive BLH anomalies are observed. This condition is conducive to the diffusion of air particulates (Fig. 5d). In the SSW onset period for split events, weak positive anomalies develop over Japan at 500 hPa, corresponding to the southerly anomalies in the BTH region (Fig. 3e). The southerly anomalies are observed in the lower troposphere (Fig. 5e). Warm anomalies occur in BTH region (Fig. 4e), and negative boundary layer height anomalies form in the BTH region, which is conducive to the accumulation of air particulates in the near surface layer (Fig. 5e). In SSW decaying period for split events, the height anomalies in the BTH region at 500 hPa diminish (Fig. 3f). However, as the cold anomalies develop in the BTH area (Fig. 4f), the boundary layer height rises again (Fig. 5f), which is consistent with the strong downward propagation of the stratospheric anomalies for split events.

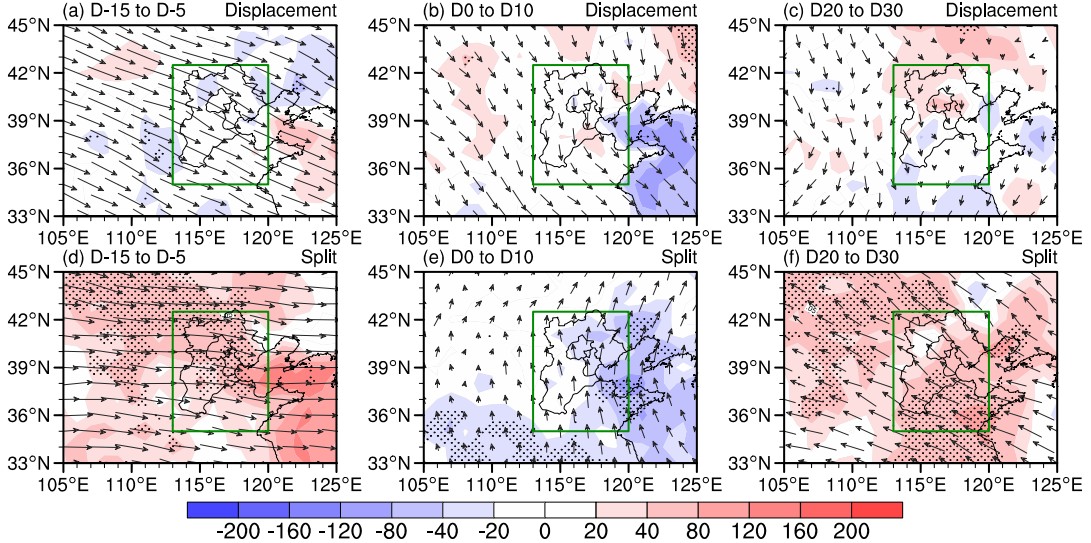

**Figure 5: Composite planetary boundary layer height (BLH; shadings) and 850-hPa wind (vector) anomalies in the BTH region during three periods of the SSW for vortex displacement events (top row) and vortex split events (bottom row). (a, d) Day -15 to day -5 in the pre-SSW period. (b, e) Day 0 to day 10 in the SSW onset period and afterwards. (c, f) Day 20 to day 30 in the SSW decaying period. The dotted regions mark the composite BLH anomalies at the 90% confidence level based on the two-sided Student's *t*-test. The green box marks the focused BTH region.**

## 6. Atmospheric environment in BTH region during two types of SSWs

The evolutions of the stratospheric circulation and the local meteorological conditions have been analyzed during three periods for the two types of SSWs. Next, we will show the corresponding evolutions of the atmospheric environment in the BTH region.

### 6.1. Daily minimum visibility in the BTH region

The $PM_{2.5}$ concentration in China is not a standard observational variable before 2013, so a composite of the atmospheric environment using the air particulate concentration is unavailable (Fan et al., 2021). The daily minimum visibility is a substitute variable for the $PM_{2.5}$ concentration to measure the air quality in the BTH region, which is extracted from the historical meteorological observations. The composite evolutions of the daily minimum visibility during the displacement and split SSW events are shown in Fig. 6 for Beijing and Tianjin (note that the minimum visibility from the historical meteorological data in Hebei stations are missing in several years, and we exclude stations from Hebei). During the two types of SSW events, the minimum visibility exhibits a consistent change. The visibility during the early stage (day -20 to day -1) of the SSW is far, denoting a clean sky and pleasant atmospheric environment. During the SSW onset and afterward (day 0 to day 20), the minimum visibility decreases and the air particulate concentration might increase. In the decaying period of the SSWs (day 21 to day 40), the visibility in the cities of Beijing and Tianjin further increases again, indicating a clean atmospheric environment and a decrease in the air particulates, which might be due to the downward propagation of stratospheric negative NAM signal.

Comparing the minimum visibility changes during two types of SSWs, it can also be found that changes in the minimum visibility are larger for split SSW events than for displacement SSW events. The mean minimum visibility of Beijing is 9.43 km with the 95% confidence interval between 8.41 and 10.44 km using the bootstrap method, and meanwhile Tianjin is 8.69 km on average with the 95% confidence interval between 7.8 and 9.59 km before displacement SSWs. The mean value in Beijing is 10.9 km with the 95% confidence interval between 9.84 and 12.04 km, and the mean is 9.13 km with the 95% confidence interval between 8.53 and 9.74 km in Tianjin before the split SSWs. During the SSW onset and afterward, the minimum visibility in Beijing is 9.16 km with the 95% confidence interval between 8.53 and 9.79 km, and in Tianjin the mean is 8.19 km with the 95% confidence interval between 7.54 and 8.84 km for displacement SSWs. In contrast, the mean decreases to 7.45 km with the 95% confidence interval between 6.65 and 8.25 km in Beijing, and the value is 7.37 km with the 95% confidence interval from 6.8 to 7.95 km in Tianjin for split SSWs. In the decaying period, the mean minimum visibility in Beijing is 9.88 km with the 95% confidence interval from 8.95 to 10.81 km, and in Tianjin the value is 8.67 km with 95% confidence interval from 8.14 to 9.21 km for the displacement SSWs. The value is 9.5 km with the 95% confidence interval from 8.59 to 10.41 km in Beijing and 8.5 km with the 95% confidence interval from 7.86 to 9.13 km in Tianjin for split SSWs. Larger change of the minimum visibility in Beijing and Tianjin during the split SSW is consistent with the stronger intensity for split SSW events and its deeper downward influence on the troposphere.

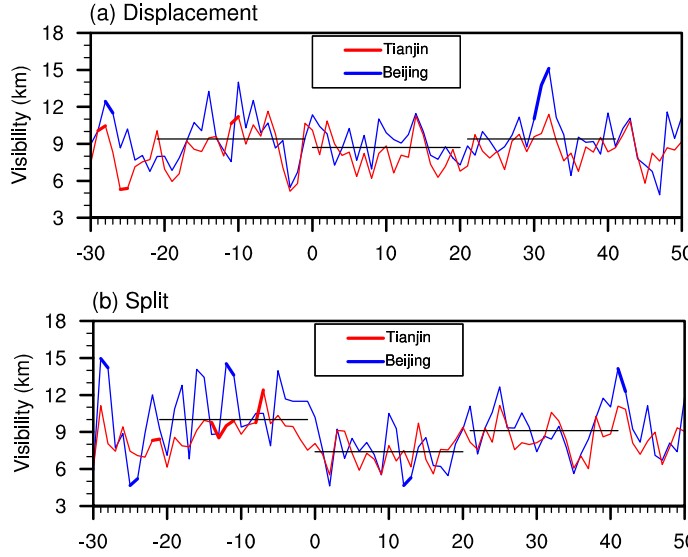

**Figure 6: Composite evolution of the minimum visibility in Beijing and Tianjin during (a) polar vortex displacement SSW events and (b) polar vortex split SSW events. The blue curve denotes the time series of the minimum visibility in Beijing while the red curve denotes that in Tianjin. The thickened curves denote that the composite is significant at the 90% confidence level using the two-sided Student's *t*-test.**

## 6.2. Haze days, fog days, and light fog days in the BTH region

Change in the minimum visibility might not be all caused by light atmospheric pollution, floating water in the air can also decrease the visibility. The composite means of the haze days, fog days and light fog days extracted from the meteorological surface observation data during the two types of SSWs are shown in Fig. 7. For both displacement and split SSWs, the number of haze days in Beijing, Tianjin, and Shijiazhuang (the provincial capital of Hebei) is relatively smaller in the pre-SSW and SSW decaying periods (P1 and P3). In contrast, the haze days increases during the SSW onset period when the downward-propagating stratospheric anomalies do not reach the troposphere. Among the three cities, Shijiazhuang has the largest mean haze days during the SSW onset stage (P2): 6 haze days for displacement SSW events and ~8 days for split SSW events. The mean haze days in Shijiazhuang during the other two periods are less: 4–4.5 days in the P1 periods and 5.5 days in the P3 period. Tianjin has the smallest mean haze days in all periods: 0.5 days in the P1 period for both types of SSWs, 1.5–2.5 days in the P2 period, and 1–1.5 days in the P3 period. The number of haze days in Beijing falls between Tianjin and Shijiazhuang, and change trends from P1 to P2 (increasing) or from P2 to P3 (decreasing) is similar among the three cities (Fig. 7a and d).

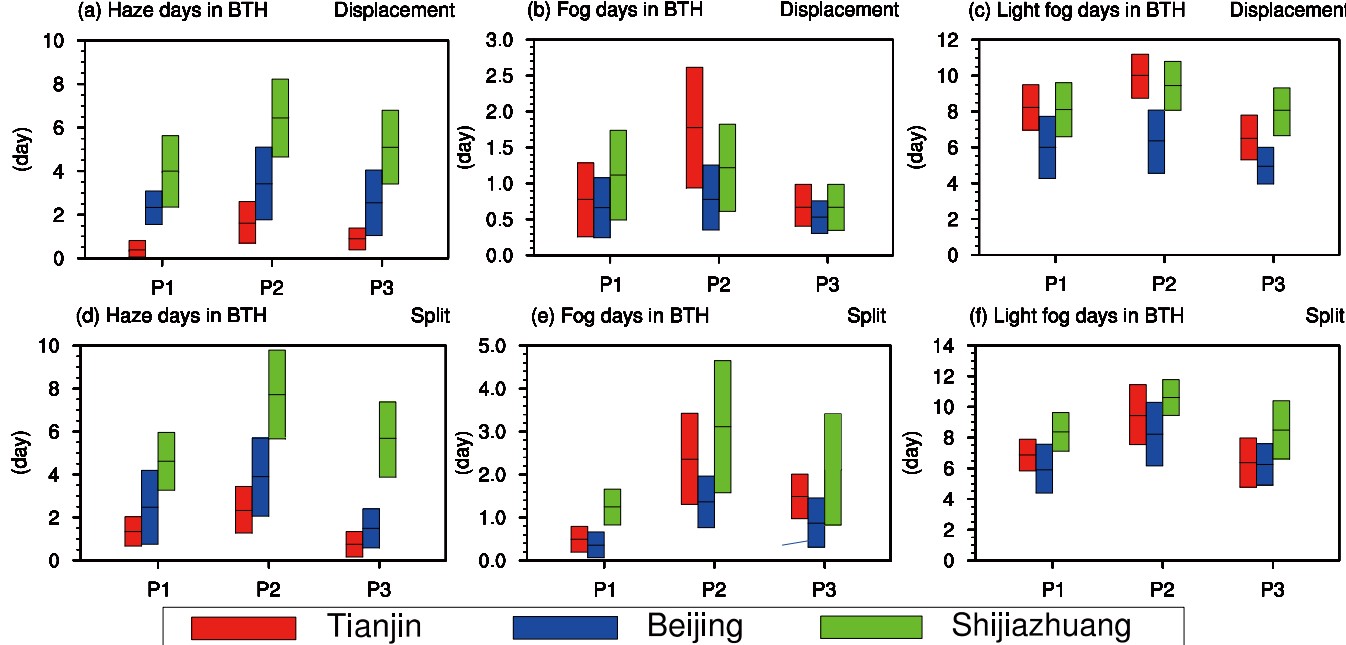

**Figure 7: Statistics of three environmental metrics during the three periods of displacement SSWs (top row) and split SSWs (bottom row). The middle line is the average, and the upper and lower limits are the 95% confidence intervals based on the bootstrap methods by resampling 1000 times. P1 denotes the pre-SSW period from day -20 to day -1, P2 denote the SSW onset period from day 0 to day 20, and P3 denote the SSW decaying period from day 21 to day 40. Composite means in three cities are shown for mean numbers of (a, d) haze days, (b, e) fog days (visibility < 1 km) and (c, f) light fog days (visibility < 10 km).**

The fog days show similar subseasonal variability during the both types of SSWs (Fig. 7b, e). Specifically, the number of fog days in BTH region is the largest during the SSW onset stage (P2) when the stratospheric anomalies do not propagate downward to the troposphere. The number of fog days are relatively less in the pre-SSW period and the SSW decaying period (P1 and P3) (Fig. 7b and 7e). In the pre-SSW period, the number of fog days for split SSWs and displacement SSWs

in the BTH region are comparable for all three cities (0.7, 0.8, and 1.1 days vs 0.4, 0.5, and 1.2 days). In the onset period of displacement SSWs, the mean numbers of fog days are ~0.8, 2, and 1.2 days in Beijing, Tianjin, and Shijiazhuang, respectively. In the onset period of split SSWs, the numbers of fog days are correspondingly ~1.5, 2.5, and 3 days in Beijing, Tianjin, and Shijiazhuang. That is, the number of fog days during the split SSW onsets in the BTH region is larger than that during the displacement SSW onsets (P2). In the SSW decaying period, the number of fog days decreases for both

displacement and split events: <1 days for displacement SSWs and 1–2 days for split SSWs. The fog days in this period are also more for split SSW events than displacement SSW events in the BTH region. Similar conclusions are also applicable to the number of light fog days during two different types of SSW events (Fig. 7c, f). Tianjin has the largest number of light fog days among the three cities considered, which might be owing to the fact that Tianjin is a coastal city and the water vapor is more abundant. In general, the number of light fog days in the BTH region is larger during the SSW onset stage (P2) than

other two periods (P1 and P3) (Fig. 7c and 7f).

## 6.3. Aerosol optical depth in the BTH region

Aerosol optical depth (AOD) is a variable measuring the optical property of aerosols and reflects the atmospheric turbidity. It represents the light transmittance on the vertical gas column with the section of a unit area (Mei et al., 2018). The AOD value falls between 0 and 1 and is dimensionless. Generally, high AOD value indicates an increase in the aerosol accumulation of the atmospheric column, which leads to a reduction of atmospheric visibility. Since the $PM_{2.5}$ concentration in China has only been systematically observed in recent years since 2013, the historical AOD data remotely sensed by NASA satellites might be used to verify the global air particulate concentration (Ou et al., 2022). The distribution of AOD anomalies in the BTH region is shown in Fig. 8 during vortex displacement and split SSW events. In the pre-SSW period for both displacement and split events (P1), negative AOD anomalies are observed in the BTH region (Fig. 8a and d), which corresponds to the high minimum visibility and a cleaning period of atmospheric environment. It is noticed that the most significant anomalies develop in the north, which might be related to circulation changes in higher latitudes during the northern winter.

Positive AOD anomalies are observed around the SSW onset period for both displacement and split period (Fig. 8b and e) when the circulation anomalies mainly develop in the stratosphere (see Fig. 1). The positive AOD anomalies during the P2 period are larger for split SSWs than displacement SSWs with the maximum positive anomalies (0.06 vs ~0.08) in the southern part of the BTH region. The positive AOD in the BTH region and the southern neighboring areas indicates that the local atmospheric quality might worsen, and the potential air pollution during split SSWs are more series than during displacement SSWs.

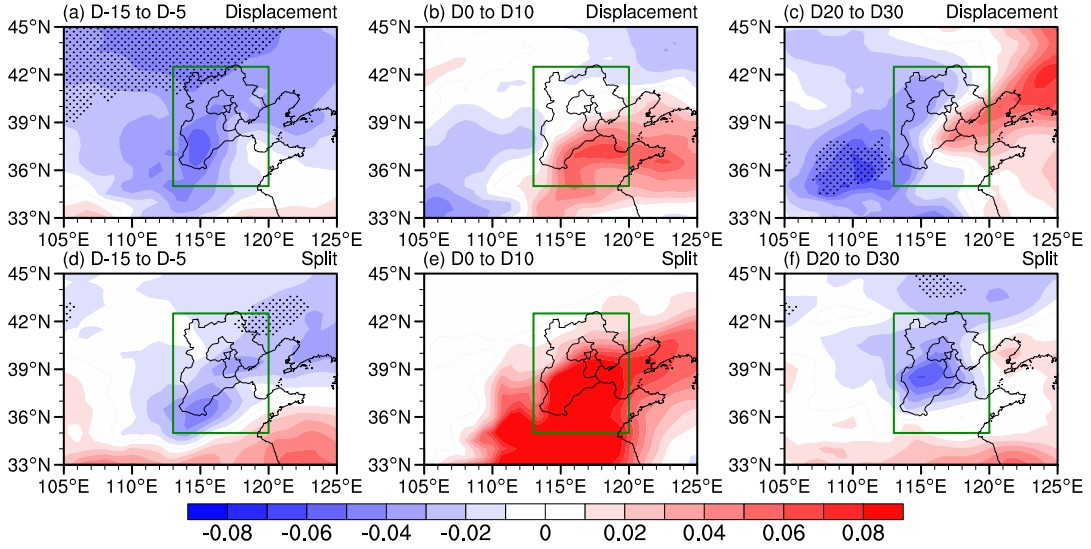

**Figure 8: Composite analysis of the aerosol optical depth (AOD) anomalies in BTH the region during three periods of the SSW for vortex displacement events (top row) and vortex split events (bottom row). (a, d) Day -15 to day -5 in the pre-SSW period. (b, e) Day 0 to day 10 in the SSW onset period and afterwards. (c, f) Day 20 to day 30 in the SSW decaying period. The dotted regions mark the composite AOD anomalies at the 90% confidence level based on the two-sided Student's *t*-test. The green box marks the focused BTH region.**

Negative AOD anomalies are observed again in the decaying period (P3) for both displacement and split SSWs (Fig. 8c and f). The minimum center is located differently: the negative AOD center (the minimum value: -0.06) is over the southern neighboring area in Central China for displacement SSWs, while the negative center (the minimum value: -0.04) is just situated over the southern part of the Hebei province. The cleaning effect during this period (P3) is consistent with the fact that the split SSWs signal in Figure 1 has a deeper downward propagation and a stronger impact on the near surface.

## 7. Summary

Using the ERA5 reanalysis data, surface meteorological observation data in the BTH region, and aerosol optical depth (AOD) from the MERRA2 reanalysis, the subseasonal evolutions of air environment in the BTH region and related changes of stratospheric and tropospheric circulation during displacement and split SSW events are systematically analyzed, respectively. The main conclusions are as follows.

i.     Seventeen major SSW events (the marginal ones have been removed from our analysis) are identified in the Northern Hemisphere from 1981–2021, including 8 displacement SSWs and 9 split SSWs. On average, the composite circumpolar easterly anomalies can persist for 45 days (day -10 to day 35) for displacement SSWs, while the composite easterly anomalies can persist for 65 days (day -15 to day 50) for split SSWs. Namely, the mean duration for split SSWs is longer than that for displacement SSWs. The stratospheric circulation anomalies associated with displacement

SSWs can propagate downward to 500–200 hPa, while the stratospheric signals for split SSWs can propagate further downward to lower levels (Fig. 1b, e). In other words, the split SSW events change more violently, and the stratospheric signal propagation is overall more evident.

ii.    As measurements for the atmospheric environment and air quality, the minimum visibility, haze days, fog days, light fog days, and aerosol optical depth (AOD) in the BTH region show consistent subseasonal changes during both

displacement and split SSWs. Specifically, the atmosphere in the pre-SSW period is clean with far visibility and fewer haze and fog days. In the SSW onset period when the stratospheric circulation anomalies do not propagate downward to the near surface, the air quality seems to worsen with less remote visibility and more haze and fog days. In the SSW decaying period when the stratospheric disturbance signal has reached the lower troposphere, the air quality in the BTH area improves with far visibility, and fewer haze and fog days again. By comparison, the subseasonal variation of the

atmospheric environment in the BTH region is more evident and robust during split SSWs events than displacement SSWs, which is consistent with the stronger change in the circulation, longer duration, and further downward propagation of the stratospheric signals for the former than the latter.

iii.   In the pre-SSW periods for both displacement and split SSW events, some differences are also noticeable. The displacement SSW events are dominated by the wavenumber 1 anomaly pattern from the troposphere to the

stratosphere, while the split SSW events are dominated by the wavenumber 2 anomaly pattern. The cold anomalies in Asia might indicate the strengthening of the East Asian winter monsoon during the pre-SSW period, and the

strengthened winter monsoon is stronger for split events than for displacement events. In the SSW onset period, the warm temperature anomalies in Asia indicate a weakening period of winter monsoon for both displacement and split SSWs, although the circulation anomalies are larger for split SSWs than displacement SSWs. In the SSW decaying period, larger cold anomalies develop in east Asia for split SSWs than displacement SSWs, possibly due to the further downward propagation of the stratospheric disturbance signals for the former than the latter. It is also observed that the cold anomalies appear in North America after the SSW onset for both displacement and split events (Cao et al., 2019; Rao et al., 2021b), and the cold anomalies in North America are also larger for split SSWs than displacement SSWs.

iv. With the gradual evolution of the large-scale circulation from the stratosphere to the troposphere during the SSW events, the boundary layer conditions in the BTH region have also changed accordingly. The boundary layer height in the BTH region shows positive, negative, and then positive anomalies from the pre-SSW period, to the SSW onset period, and then to the SSW decaying period, respectively. Change in the local meteorological conditions are also consistent with the subseasonal variation of the local air quality in the BTH region. In contrast, the variation of the boundary layer height during displacement SSWs do not necessarily cooperate with the large-scale circulation from the stratosphere to the troposphere to modulate the atmospheric environment in the BTH region. It can be inferred that the stratospheric pulse signal with further downward propagation might affect the local meteorological conditions for particulate diffusion and dilution.

With China's strict implementation of energy conservation and emission reduction policies in the past 10 years, the air quality and atmospheric environment have improved significantly (Ding et al., 2019). However, the impact of changes in meteorological conditions on air quality is still very evident. Compared to previous studies emphasizing the possible impact of tropospheric meteorological conditions and teleconnections on the air quality in the BTH region (Yin and Wang, 2017; Huang et al., 2018; Zhai et al., 2019), our study further points out some typical stratospheric disturbance such as SSWs can also modulate the subseasonal variability of the air quality in one of the most populated regions across China. Considering that the stratospheric circulation anomalies can lead the tropospheric circulation anomalies by days to weeks, an improved predictive skill might become possible after the stratospheric signals are also considered in the subseasonal atmospheric environment forecast systems. With abundant SSW samples, our study has also confirmed the generalization of the results based on individual case studies (Lu et al., 2021a, 2022). The composite results in this work are consistent with previous case studies, although only 8 or 9 SSWs are considered. In the future, outputs from climate-chemistry coupled models (e.g., Liang et al., 2022; Rao et al., 2022) can be used to further improve the robustness of diagnostic results with a much larger SSW sample size.

**Acknowledgements**

This work was supported by the National Natural Science Foundation of China (Grant Nos. 42088101 and 42175069), and the National Key R&D Program of China (2018YFC1505602).

**Data Availability Statement**

The ECMWF provides the ERA5 reanalysis (https://cds.climate.copernicus.eu/). The NASA provides the MERRA2 atmospheric reanalysis (https://disc.gsfc.nasa.gov/datasets). The daily observation data is from the China Meteorological Information Center (http://data.cma.cn/).

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
