# Peer review of "Possible influence of Sudden Stratospheric Warmings on the atmospheric environment in the Beijing-Tianjin-Hebei region"

_Atmospheric Chemistry and Physics, 2022_

## Author Comment (AC1)

**Response to Reviewer # 1**

**(1) General comments:**

The manuscript by Lu et al., submitted to ACP, presents observationally-based evidence of Sudden Stratospheric Warmings (SSWs) influence on air quality (AQ) over the Beijing-Tianjin-Hebei (BTH) region. The authors use reanalyses and observations fields over the 1980-2021 period to analyse stratosphere-troposphere coupling during 17 SSWs events, distinguishing between displacement (8) and split (9) type events. Responses of surface AQ over the BTH region, albeit substantially limited by the number of SSWs and (likely) large natural variability, show consistent changes with regard to current stratosphere-troposphere coupling understanding. This work extends and supports previous studies findings by including a number of SSW events, and finds stronger AQ responses over the BTH region associated with split SSW events.

The present study is of interest to the atmospheric chemistry and physics community. However, the language of the manuscript could be substantially improved, which would help to achieve a clear logic flow. The aim of the paper is clearly defined and technical concepts are usually presented. The methodology is sensible; however, it lacks a more detailed description to clearly understand the significance of the findings and their implications (see below). Results are well presented and support the conclusions. However, the analysis can greatly benefit from an additional discussion about the robustness of the results, e.g., limitations due to a relatively small number of SSW events and how this could be addressed in future work. I would recommend the present study for publication after the comments below are addressed.

Response: Thank you for your comments. We have included all of your suggestions in our revision, which help to improve the overall quality of our manuscripts.

**(2) Specific comments:**

The manuscript would greatly benefit from a substantial language revision (i.e., technical). The manuscript in its current form makes difficult the reading and understanding of the analysis. For example, please avoid using parenthesis to highlight the opposite (i.e., it takes at least two readings of the sentence to understand the meaning).

Response: The author has made a lot of technical revisions to the language of the manuscript.

We avoid using parenthesis to state the opposite position this time.

"According to the principle of geostrophic winds, decelerated westerly jets in the circumpolar region are accompanied with rise of polar cap height and/or the reduce of

the midlatitude heights, and accelerated winds are accompanied with decrease of polar height and/or rise of midlatitude heights." (L189-192)

The statistical test (i.e., t-test) used to define the significance of the results needs to be more clearly described. This involves the assumptions made, such as the distribution of the samples (e.g., normal or Student's t), and the variance of the samples (e.g., equal?), as well as the test used (i.e., one-sided or two-sided). All these details will greatly influence the statistical significance of the findings (e.g., Krzywinski & Altman, 2013; doi:10.0.4.14/nmeth.2698).

Response: The statistical test used to define the significance of the results has been clearly described this time.

"One-sample two-sided *t*-test is to test whether the difference between a sample average and a known population average is significant. When the population distribution is normal, but the samples are not large, the deviation of the sample mean from the population mean show a *t*-distribution. As the SSW samples are limited, it is reasonably to use the Student's *t*-test for this study. The one-sample *t*-test is calculated as $t = \frac{\bar{x}-\mu}{\frac{\sigma_x}{\sqrt{n}}}$, where *n* is the sample number, $\bar{x}$ is the sample mean, and $\sigma_x$ is the sample standard deviation. The null hypothesis is that the *t*-value is zero (i.e., $\bar{x}=\mu$) if the sample mean shows insignificant difference from the population mean. Otherwise, the sample mean is significant different from the population mean if the null hypothesis is rejected (Krzywinski and Altman, 2013). In order to test the credibility and consistency of data, the bootstrap method is adopted to calculate the confidence level (e.g., Alfons et al., 2022) for the mean visibility, haze days, fog days, light fog days by resampling 1000 times with a sample size proportion of 0.5 for both displacement and split SSWs." (L137-146)

The authors often describe their results without addressing whether the signals are significant or not (i.e., yet significance is provided in their analysis). It is important to focus on relevant features (i.e., statistically significant) to understand the implications of the results.

Response: We considered your suggestion and mentioned the significance level in several places.

- "However, only the easterly anomalies above 50 hPa from day -10 to day 20 are statistically significant and reach the maximum (-30 m/s) around the onset dates (Fig. 1b)" (L161-162)

- "However, only the easterly anomalies above 200 hPa from day -15 to day 40 are statistically significant (Fig. 1e)." (L165-166)

- "No significant zonal-mean temperature anomalies are observed in the troposphere…" (L180)

- "In contrast, for the split SSWs, the positive geopotential height anomalies appear since day -22 and propagate downward to the troposphere instantly, where significant signals are observed." (L198-199)
- "…and the significant positive height center is located around the Bering Strait and North Pacific (Fig. 2a)." (L215-216)

Please see our revised manuscript for more details.

The manuscript lacks a discussion about the robustness and limitations of the results, e.g., taking into account that only 8-9 events for displacements and split types are accounted, respectively; what else can be done in future work to gain confidence?

Response: We added some discussion this time. Thank you very much!

"The composite results in this work are consistent with previous case studies, although only 8 or 9 SSWs are considered. In the future, outputs from climate-chemistry coupled models (e.g., Liang et al., 2022; Rao et al., 2022) can be used to further improve the robustness of diagnostic results with a much larger SSW sample size." (L502-505)

**(3) Technical corrections:**

L30. References only refer to AQ health impacts. Please, include/expand references that support haze pollution impacts on the ecological environment, transportation, and so forth.

Response: A reference about the impact of haze pollution on the ecological environment has been added. (L31)

L35-40. What is the Arctic Oscillation? Please describe (e.g., one sentence).

Response: "During the AO positive phase, the low pressure in the Arctic region deepens, and the high pressure in the midlatitudes intensifies, limiting the southward expansion of the cold air in the polar region.". (L41-43)

L63-64. Since this is a key feature of SSWs events, the authors may want to move the sentence at an earlier stage of the paragraph, i.e., "SSW event is a typical phenomenon of two-way coupling between stratosphere and troposphere (Hu et al., 2014)".

Response: The sentence has been moved to an earlier state of the paragraph. (L51-52)

L104-106. Observations provided by the China Meteorological Information Center need reference and/or link.

Response: Revised. (L109)

L137-141. Please clarify the meaning of the dates provided, e.g., onset of the SSWs events?

Response: Added "…and their onset dates are as follows". (L151, 153)

L148 and L152-154. Only for the split SSWs cases, right? With the analysis being not statistically significant below ~200hPa.

Response: Considering the comments from the other reviewer, the composite for all SSWs is removed.

- "When the displacement SSWs are considered, the zonal mean zonal wind anomalies at 60°N only appear above 200 hPa and do not propagate downward to lower troposphere. The easterly anomalies begin to appear 10 days before the onset dates and last until day 35. However, only the easterly anomalies above 50 hPa from day -10 to day 20 are statistically significant and reach the maximum (-30 m/s) around the onset dates (Fig. 1b)." (159-162)

- "In contrast, the easterly anomalies begin to appear since day -15 for split SSWs, and the wind anomaly magnitude is also stronger, reaching the maximum intensity (-35 m/s) several days before the SSW onset (Fig. 1e). The easterly anomaly signal can last until day 50, which propagate downward to the near surface since day 20. However, only the easterly anomalies above 200 hPa from day -15 to day 40 are statistically significant." (L162-166)

L166. Warmest anomalies for SSWs displacement events appear on day 5? Do you mean day -5?

Response: Changed. (L175)

L169-171. Such statement needs reference.

Response: Added a reference here. (L182)

L188-189. It is not clear what the sentence is referring to. Please clarify.

Response: Added "…for the split SSWs…" (L198)

L194-196. "... might have a downward influence on the tropospheric circulation. ..." This is a bit confusing given the statement in L199-200 ("... the stratosphere-troposphere coupling in the extratropics, which is followed by discernable tropospheric circulation anomalies...").

Response: Changed to "…which has a downward influence on the tropospheric circulation anomalies." (L204-205)

L200-201. I do not understand this sentence here. Are the authors not focusing on describing stratospheric circulation anomalies?

Response: Deleted "and troposphere". (L211)

L233. PNA has not been introduced.

Response: Added. (L256-258)

L250. "The enhanced wavenumber 2 can also propagate upward into the stratosphere and split the polar vortex" needs a reference.

Response: Added. (L278)

L257. "are shown Fig. 4" -> "are shown in Fig. 4"

Response: Changed. (L283)

L284-287. The SSWs stratosphere-troposphere coupling needs reference.

Response: Added two references here. (L314)

Figure 7. The authors may want to include confidence intervals of the environmental metrics shown in the figure.

Response: Added. (L401-405)

L423. "... is more significant..." -> "... stronger..."?

Response: Changed. (L477)

L440. "Compared with the some previous..." -> "Compared to previous..."

Response: Changed. (L495)

---

## Author Comment (AC2)

**Response to Reviewer # 2**

Lu et al. studied how stratospheric sudden warming (SSW) events can influence the air quality in Beijing-Tianjin-Hebei region in east Asia using a combination of re-analysis and observational datasets. In particular they study a coupling on subseasonal-to-seasonal time scales and a distinction between split and displacement SSWs. Their work is based on an interesting research question and the corresponding results could indeed be useful for a wide audience. The paper generally follows a reasonable structure and covers a range of analyses covering large and small scale diagnostics in both stratosphere and troposphere. The figures are mostly easy to understand and presented in an overall adequate way. The language used sometimes seems quite cumbersome or unnecessarily complicated.

However, I feel like the content of the paper could be more focused on the new aspects of the study, e.g., by emphasising in more detail the concrete chain of processes involved in the coupling. I am further not fully convinced by the robustness of some of the signals and further discussion and/or analysis is required. In my opinion, the manuscript needs some substantial revision (see comments below), but could afterwards very well form a valuable contribution to current research.

Response: Thank you very much for your valuable comments and encouragement. We considered all of your comments, which have helped to improve the quality of our manuscripts.

**General comments:**

1. A large part of the manuscript covers the influence of SSWs on the (large scale) tropospheric circulation, which is a topic well discussed in the existing literature. On top, it seems to me like much of the corresponding results are not very convincing: Fig. 1, for example, is supposed to show differences in the dynamical downward coupling between split and displacement SSWs in several metrics, but one can hardly see any significant or substantial anomalies near the surface in any of the panels. This might simply be an unlucky choice of diagnostics. A possible approach here might be to extend the literature review in terms of SSW-research rather than "reinventing the wheel" and instead focus more on the connection between large scale circulation and regional circulation/pollution.

Response: The introduction introduces the differences between displacement and split SSWs. Because the air pollution shows a strong seasonal dependence in some regions, this paper only selects SSW events in midwinter to diminish the interference from the seasonal cycle. Further, we rearranged Figure 1 to well address this concern. The composite evolution for all SSWs was removed, and only displacement and split SSWs are shown. We focused more on the comparison between split and displacement SSW events. (L157-207)

The chain in this article is that: (1) SSW affects the large-scale circulation in the troposphere through stratospheric-tropospheric coupling, (2) the local boundary layer meteorological conditions in Beijing-Tianjin-Hebei region is modified, (3) the local atmospheric environment is modified.

2. One of the main points of the study is the distinction between split and displacement events, however, none of the figures show specific difference-plots. Hence, small differences can be masked by large absolute anomalies despite still being important. E.g., Figs. 2b and e or Figs. 6a and b could be easier to interpret if you also show corresponding differences.

Response: After careful consideration, the difference was added for Fig. 2. The difference in Fig. 2 is large and significant. We added the split minus displacement difference.

- "In the SSW onset and decaying periods for both displacement and split SSWs, a negative NAM is observed. To clearly reveal the difference between the split and displacement SSWs, the split minus displacement composite is also shown (Fig. 2g–i). The composite in the pre-SSW period shows that the difference is largest and most significant in the North Atlantic (Fig. 2g). In the SSW onset period, this composite difference resembles a wavenumber-1 like pattern, which denotes a stronger wave-1 forcing for displacements than splits (Fig. 2h). In the post-SSW period, the difference is still evident over the Bering Strait and North Pacific (Fig. 2i)." (L239-244)

- "The positive height center over the Arctic for split SSWs is more inclined to the Iceland and Greenland, whereas this center for displacement SSWs is round over the North Pole." (L248-249)

3. You should probably extend your discussion about robustness and significance of your results or even extend your analysis to extract more significant signals. Fig. 6 suggests reduced visibility following the first two weeks after split events both overall and compared to displacement events. However, the diagnostic shows high day-to-day variability and I am not fully convinced this plot shows an actual downward influence. The same holds for other figures. One way you could deal with these weakly significant signals is to further emphasise consistency between diagnostics.

Response: In order to increase the credibility and consistency of data, the mean visibility value and 95% confidence interval are estimated based on the bootstrap method by resampling 1000 times for both displacement and split SSWs. Table R1 and Table R2 are shown exclusively for your reference. Figure 7 also adds the 95% confidence interval.

This subsection was revised substantially. (L367-381)

**Table R1.** The mean visibility value with its 95% confidence interval estimated using the bootstrap method by resampling 1000 times for displacement SSWs (unit: km).

| | P1(pre-SSW) | P2(SSW onset) | P3(post-SSW) |
|---|---|---|---|

| | average | 95% CI | average | 95% CI | average | 95% CI |
|---|---|---|---|---|---|---|
| Beijing | 9.43 | (8.41,10.44) | 9.16 | (8.53, 9.79) | 9.88 | (8.95,10.81) |
| Tianjin | 8.69 | (7.8, 9.59) | 8.19 | (7.54, 8.84) | 8.67 | (8.14, 9.21) |

**Table 2.** The mean visibility value with its 95% confidence interval estimated using the bootstrap method by resampling 1000 times for split SSWs

| | P1 (pre-SSW) | | P2 (SSW onset) | | P3(post-SSW) | |
|---|---|---|---|---|---|---|
| | average | 95% CI | average | 95% CI | average | 95% CI |
| Beijing | 10.9 | (9.84,12.04) | 7.45 | (6.65,8.25) | 9.5 | (8.59,10.41) |
| Tianjin | 9.13 | (8.53, 9.74) | 7.37 | (6.8, 7.95) | 8.5 | (7.86, 9.13) |

4. I feel the chain of processes leading from a polar vortex break down to changes in regional air quality is not made clear enough. Fig. 5 shows differences in regional winds and planetary layer height patterns between split and displacement events, which are supposed to explain the changes in air quality. However, during days 0-10 (when the differences in visibility are largest according to Fig. 6) the magnitude in both are relatively equal, while for days 20-30 (when Fig. 6 suggests almost no differences) you find strong dynamical differences.

Response: A comprehensive understanding might be possible by analyzing Figs. 3–5, and the effects of large-scale tropospheric circulation evolution on pollution diffusion conditions in BTH region are revised by adding more details. (L316-343)

To well address your concern, quantitative description for Fig. 6 is added this time to well compare the difference between displacement and split SSWs. You might find the minimum visibility is indeed different between displacements and splits. (L367-381)

**Specific comments and typos:**

L13: I would move the sentence starting "As the duration of split SSW..." to later in the abstract as you should first address SSWs in general and then make the distinction between different subclasses.

Response: Added "Major SSW events are divided into polar vortex displacement SSW and polar vortex split SSW" before the sentence. (L13)

L64-66: You do not necessarily find enhanced wave forcing preceding SSWs (eg: de la Cámara, 2019, JoC)

Response: Revised.

- "Before the SSW onset for some events, the upward propagation of planetary waves from the troposphere to the stratosphere is enhanced (de La Cámara et al., 2019; Rao et al., 2019b), which might be owing to the preceding tropospheric blocking and/or

deepening of the climatological trough (Rao et al., 2018, 2020; Baldwin et al., 2021)." (L67-69)

- "Another trigger for SSWs is the stratospheric dynamics and the vortex geometry in the lowest stratosphere (de La Cámara et al., 2019)." (L69-70)

L67: Maybe make clear this particularly holds for the zonal mean anomalies!

Response: Revised. "After the SSW onset, the atmospheric zonal mean anomalies generated by SSW events …" (L70-71)

L84: Please add a note introducing $PM_{2.5}$

Response: Added "… (small particles with the aerodynamic diameter equal to or less than 2.5 μm in the atmosphere) …" (L87-88)

L101: m/s is missing for g

Response: Added "…m s$^{-2}$". (L104)

L112: 121 days seems like a large window size. are your results sensitive to it?

Response: Not really. "Daily anomalies refer to the department from this smoothed daily climatology with a window of 91 days (three months or one season) to remove the high-frequency variability. The results are unchanged if we change the window between 61 and 121 days." (L114-116)

L118: Please add at least one sentence describing what these diagnostics are.

Response: Revised. "Vortex-centric diagnostics are used to categorize the type of SSW events, which can calculate the vortex centroid latitude and longitude (Seviour et al., 2016). In addition to the vortex-centric parameters, the aspect ratio can also be calculated based on the two dimensional vortex moment diagnosis of the vortex shape, which are used to define a vortex uniquely, and an "equivalent ellipse" is defined as the representative of a vortex (Mitchell et al., 2011; Seviour et al., 2016)." (L122-126)

L135: how many minor warmings do you find? is it worth showing a plot for these events similar to Fig. 1 (in a supplement)?

Response: Minor SSWs are much more than the major SSWs on average, but they do not show a significant impact on the $PM_{2.5}$ evolution. We decide not to show. A sentence was revised. "…the SSW is usually classified as a minor event, which is excluded from our analysis." (L57-58)

L140: 2 January 2019 is listed as both displacement and split event, does it enter both composites during your analysis?

Response: The 2019 SSW is split event after the onset date. We revised this sentence. (L154)

L160: This seems to be a statement that should be clear at this point in the paper, especially because it is also mentioned in the introduction. I would much rather like to see a similarly detailed discussion on how the tropospheric circulation might affect regional air quality.

Response: We understand your concern. We should briefly compare the split and displacement SSW before we go to the impact on the regional air quality directly, which provide a background. To well address your concern, this part has been modified, with emphasis on the difference between the two SSWs. (L172-182)

L178-180: This statement also seems a bit misplaced within the results section.

Response: This statement is also based on Fig. 1. (L189-192)

L180: denote -> are consistent with

Response: Changed. (L192)

L194-195: But you just investigated this, right? So do you conclude there is a downward influence or not?

Response: Yes. We added a reference, and revised this sentence. (L204-205)

Fig. 3: I suppose the small green box marks the area of interest? Maybe make the box more pronounced and mention it in the caption. Also: the downward influence of stratospheric anomalies is relatively weak in the mid-troposphere and usually strongest near the surface (eg: Baldwin+Dunkerton, 2001, Science), so maybe 500hPa is not the best level to look at.

Response: Thank you very much. We read this reference carefully. Actually, the mid-troposphere is widely analyzed in literature especially when studying the air quality. We still keep this analysis after careful consideration. We added the focused region in the caption: "The green box marks the focused BTH region." (L228, 270, 298, 349, 443)

L404: The word "persist" can sound like you are talking about specific events that last this long; make clear you are talking about the average.

Response: Changed to "On average, the composite circumpolar easterly anomalies can persist for 45 days…" (L456-457)

L406: can propagate

Response: Changed. (L460)

L404-406: Based on your results I am not really convinced you can make this statement with such certainty. Fig. 1 shows no signal for either type of SSW below 500hPa except for U (with essentially no statistical significance). Further, it seems like the surface temperature anomalies are stronger in the displacement case (if there are any significant differences at all) following the SSW.

Response: This sentence was revised. "The stratospheric circulation anomalies associated with displacement SSWs can propagate downward to 500–200 hPa, while the stratospheric signals for split SSWs can propagate further downward to lower levels (Fig. 1b, e)." (L459-461)

L422-423: This is a hypothesis, right? You don't actually look at any "pure" monsoon diagnostic.

Response: We revised this sentence. (L475-479)

L439: Did you remove a potential inter-annual trend due to these policies or other climate signals?

Response: Yes. "Daily anomalies refer to the detrended deviation …" (L114)